# VISUAL TOPICS VIA VISUAL VOCABULARIES

## ABSTRACT

Researchers have long used topic modeling to automatically characterize themes in a text corpus without supervision. Can we extract similar structures from collections of images? To do this, we propose *visual topic modeling*, a method to analyze image datasets by decomposing images into segments, and grouping similar segments into visual "words". These vocabularies of visual "words" enable us to extract visual topics that capture hidden themes distinct from what is captured by classic unsupervised approaches. We evaluate our visual topics using standard topic modeling metrics and confirm the interpretability of our visual topics via a human study.

## 1 INTRODUCTION

The structure of written language is simple — words are comprised of letters, and documents are comprised of words. The body of words used in English, or the English *vocabulary*, allows us to create meaningful text from this fixed collection of words. This vocabulary-based structure also allows us to use certain natural language processing (NLP) techniques, like topic modeling, to understand and interpret language datasets.

Topic modeling is an unsupervised learning algorithm that extracts latent variables from large datasets in the form of topics (Vayansky & Kumar, 2020). These topics capture groups of related words within a body of text, and experts have directly interpreted topics in domains like medicine and social science (Doogan et al., 2020; Liu et al., 2016; Ramage et al., 2009) to understand and discover information within datasets. Topic modeling is an especially desirable explanation technique due to (1) the human-interpretability of topics, and (2) the uncovering of *relationships* between words in a document that are not always semantically similar. For example, "ball", "field", and "jersey" are related in that they may appear together in a "sports" topic even though these three words represent different objects and are not close in a typical word embedding space.

Given its popularity in NLP, we hypothesize that topic modeling could also discover relationships within images, where topics are distributions over related "words" in an image. These relations are distinct from image clusters, which often explicitly optimize for visual similarity. However, topic modeling algorithms process documents of words, whereas images do not have any such obvious linguistic structure. Getting topics for images in the same way we get linguistic topics first requires a vocabulary for images.

Therefore, we develop a "visual vocabulary": a mapping of similar segments into "visual words" that constitute a discrete vocabulary for image data. We then use this visual vocabulary to transform images into a document representation that is compatible with classic topic modeling algorithms. The resulting methodology is called *visual topic modeling*: a procedure for extracting topical relationships from image datasets. We demonstrate the benefit of visual topics in two ways. First, we empirically and theoretically show how visual topics uniquely capture relationships across images; these relationships differ from classic approaches for interpreting vision datasets, which tend to focus on visual similarity. Second, we generate visual topics across a suite of image datasets and show our visual topics are of high quality based on standard topic modeling metrics and human evaluation.

Our contributions are as follows:

1. We propose visual topic modeling as a way to explain hidden themes in an image dataset. Our methodology derives a visual vocabulary from images as an interface between image datasets and topic modeling algorithms.

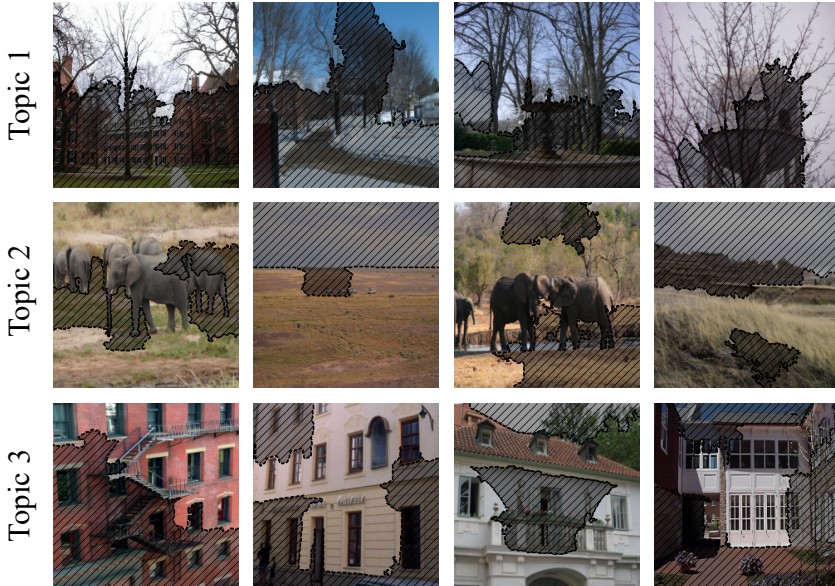

Figure 1: Examples of visual topics for the SUN397 dataset. For each topic $T$, we show four exemplar documents $D$ which maximize $P(T|D)$, as well as the corresponding segments $w$ with high $P(w|T)$. Both ($P(T|D)$ and $P(w|T)$) are outputs of the topic modeling algorithm. Topic 1 contains leafless trees in the wintertime, highlighting the bare branches in addition to the skies behind them. Topic 2 contains segments of the Savannah, along with elephants in a similar landscape. Lastly, Topic 3 shows a variety of buildings with different textures and colors, but similar window layouts.

2. We demonstrate, via experiments and a theoretical example, how visual topics capture relationships in images distinct from what existing dimensionality reduction methods capture.

3. We adopt standard topic modeling evaluations from the NLP literature to assess our visual topics. We find our topics to be of good quality according to topic modeling metrics and highly interpretable via human evaluation.

## 2 VISUAL TOPICS VIA VISUAL VOCABULARIES

Based on its wide-reaching success in NLP, topic modeling can potentially capture meaningful relationships between components of images. However, there is a type-mismatch: images in vision are collections of pixels, whereas topic modeling algorithms expect documents consisting of sequences of words. In order to use topic modeling on image datasets, we must first transform images into documents of "visual words" to interface with topic modeling algorithms.

We formalize this problem as finding a mapping $h : \mathcal{X} \rightarrow \mathcal{V}^m$ from images $\mathcal{X}$ to documents $\mathcal{V}^m$, where documents are a sequence of $m$ words from a discrete vocabulary $\mathcal{V}$. Once we have mapped our examples into documents with $h$, we can apply standard topic modeling algorithms. But how exactly should we map images into documents? As a naive approach, consider implementing $h$ directly with an image-to-text model, such as CLIP, Radford et al. (2021) to directly create a caption for each image. However, captioning models may fail to translate all of the key components of an image into text (Tong et al., 2023).

We instead propose a two-step mapping that preserves all of the components: we first partition an image into a list of $m$ segments with a segmenter $s : \mathcal{X} \rightarrow \mathcal{X}^m$, where $s(x) = (z_1, \ldots, z_m)$ for segments $z_j \in \mathcal{X}$. To transform these segments into visual words, we define $v : \mathcal{X} \rightarrow \mathcal{V}$ which maps each image segment to a word from a discrete vocabulary $\mathcal{V}$. Then, we can define our visual document generator $h$ as the following:

$$h(x) = [v(s(x)_1), \ldots, v(s(x)_m)] \tag{1}$$

In summary, we break an image into segments and map each segment to a word in a discrete visual vocabulary. Each image's corresponding words are then concatenated together to create a document.

---

**Algorithm 1** Visual Topic Modeling

---

**Require:** Images $x_1, x_2, \ldots, x_n$, Segmenter $s$, Embedding Model $f$
**Require:** Integers $K, T$ where $K \leftarrow$ num clusters, $T \leftarrow$ num topics

> **for** $i = 1 \ldots n$ **do**                           ▷ Segmenting
>      $s_i \leftarrow s(x_i)$
> **end for**
> $c_1, c_2, \ldots, c_K \leftarrow k\text{-means}(\bigcup_i^n f(s_i))$          ▷ Clustering
> **for** $i = 1 \ldots n$ **do**                  ▷ Document Construction
>      **for** $j = 1 \ldots |s_i|$ **do**
>          $d_{ij} \leftarrow \text{argmin}_{l=1\ldots K} \|f(s_{ij}) - f(c_l)\|_2^2$
>      **end for**
> **end for**
> $t_1, t_2, \ldots, t_T \leftarrow \text{LDA}(d_1, d_2, \ldots, d_n)$      ▷ Visual Topic Modeling

---

These documents can then be fed into a topic modeling algorithm such as Latent Dirichlet Allocation (LDA). The whole approach, which we call visual topic modeling, is detailed in Algorithm 1.

**Creating a Visual Vocabulary.** While there exist standard segmentation methods for $s$, we need to create a suitable vocabulary generator $v$ for vision. A key part of our approach is to therefore construct a discrete, visual vocabulary $\mathcal{V}$, and define a mapping $v$ from segments to $\mathcal{V}$. After segmenting our image into segments $(z_1, \ldots, z_m) = s(x)$, we use an image embedding model $f$ from Dosovitskiy et al. (2021) to extract embeddings for each unique segment. We then cluster the resulting segment embeddings into $K$ clusters $(c_1, \ldots, c_K)$ using $k$-means. Each of the $K$ clusters then is taken to be a discrete *word* in the visual vocabulary, resulting in a vocabulary of size $|V| = K$. Lastly, we assign each segment to its nearest cluster in embedding space to map segments to words. This procedure results in the following vocabulary generator:

$$v(z) = \underset{l \in \{1, \ldots k\}}{\arg\min} \|f(z) - f(c_l)\|_2^2 \tag{2}$$

Note that this vocabulary is not the English language—at a first glance, it may appear to be a non-sensical collection of cluster labels. However, this vocabulary mimics the *structure* of language. The English language has a fixed set of words that appear across sentences in a text corpus. Similarly, the visual vocabulary is a fixed set of clusters which are shared across different images. Figure 2 shows an example visual vocabulary.

**Visual Documents and Topic Modeling.** With our mapping from images to a discrete visual vocabulary, we now have a natural way to form documents. We simply concatenate the words corresponding to the segments in an image as done in Equation 1 to get a visual document (see Figure 2). With these documents, we can directly apply any topic modeling algorithm to our image dataset, creating *visual topics*. The resulting topics serve as an interpretable dimensionality reduction to explain an image dataset. In this work, we use LDA [1] as our topic modeling algorithm, given its simplicity and establishment in the field.

## 3   TOPIC MODELING UNCOVERS RELATIONS IN IMAGES

The hallmark of topic models in language is the ability to discover relations between words that constitute overall themes. For example, a political topic in language can include words such as "government", "president", and "state" even though these are distinct entities. Our visual topics can uncover similar structures in image data, with several examples shown in Figure 1. Topic 1 contains bare tree branches, clear skies, and cloudy backdrops. Crucially, these three components have different shapes and colors and are not clustered together. In this section, we analyze what makes these topic relations different from classic interpretable structures for images such as clusters.

---

[1]Latent Dirichlet Allocation (LDA) is probabilistic topic modeling technique that assumes each document in a corpus is a mixture of topics, and each topic is a distribution over words. The algorithm uses Bayesian inference to return the most likely topic-word distributions – $P(w|T)$ for all words $w$ and topics $T$, and document-topic distributions – $P(T|D)$ for all topics $T$ and documents $D$.

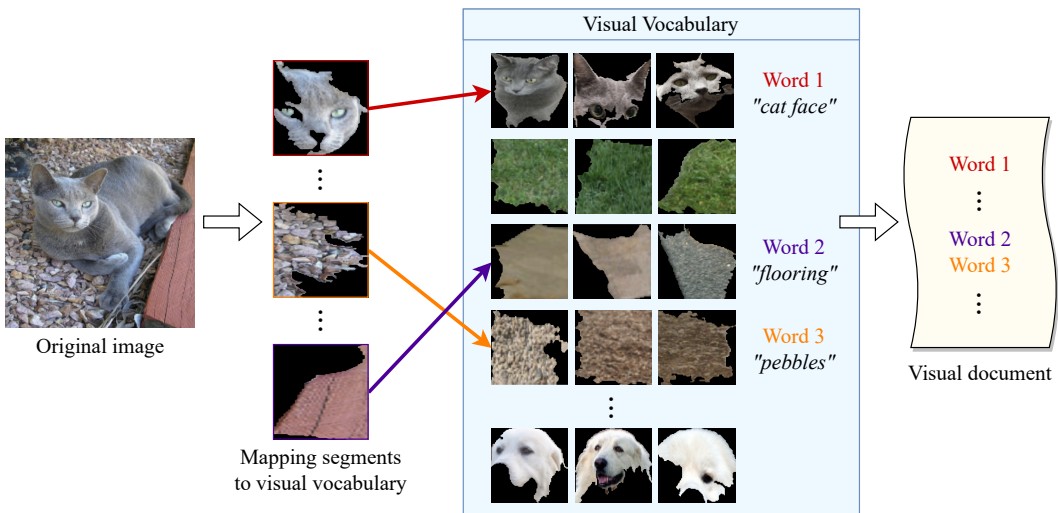

Figure 2: Mapping an image from the Pets dataset (Parkhi et al., 2012) to a visual document using our constructed visual vocabulary. We construct a discrete vocabulary for an image dataset by segmenting each image and clustering similar segments together, treating each cluster as a "word". We then map each image to a visual document by mapping each segment to its visual "word".

### 3.1 TOPIC RELATIONS FOR IMAGES ARE DIFFERENT FROM CLASSES AND CLUSTERS

Clustering has been used as the predominant approach to group and understand patterns in images (Ghorbani et al., 2019; Sambaturu et al., 2020; Bai et al., 2021), whereas topic modeling is a more popular approach in language (Jelodar et al., 2019; Vayansky & Kumar, 2020). However, clusters and topics extract fundamentally distinct structures from the underlying data.

In vision, clusters tend to contain only visually *similar* images. Clustering algorithms, by construction, explicitly optimize for groups with high similarity according to some metric. In fact, the focus on similarity is so strong that maximizing class purity of each cluster is the end-goal of modern clustering algorithms Adaloglou et al. (2023). Due to this focus on visual similarity, images in a cluster tend to be from a single class, as images from the same class are most similar. On the other hand, topics capture *related* words that co-occur together but may not be similar. For example, the words "torment" and "torrent" are both similar in character distance, but have very different meanings and are not related. Conversely, "government" and "president" are quite different in character distance, but are related political entities.

Therefore, when topic modeling an image dataset, we expect the resulting visual topics to capture different information from traditional clusters. In particular, we analyze whether topics contain structures that span across multiple classes, given traditional clusters tend to be homogeneous. If topics are heterogeneous with respect to classes, then they must contain different structures.

To measure this difference, we get visual topics for all datasets listed in Appendix A.3 and calculate the entropy of the top 50 images within each topic[2] — see Figure 3a. An entropy of 0 indicates the top images in each topic all originate from a singular class, while an entropy of 1 indicates a uniform distribution across all classes. We observe that our topics have an exceptionally high entropy, indicating they capture more than just visual similarity. The vast majority ($> 95\%$) of our topics have an entropy of 0.125 or greater.

Given that clusters of images are built to be homogeneous with respect to class, they do not capture relations that span multiple classes as visual topics do. To capture more fine-grained structure than classes, another line of work has explored clustering segments instead of the entire image (Ghorbani et al., 2019). Although Ghorbani et al. (2019) focuses on per-class clusters, segment clusters are

---

[2]For a given topic $T$, we use the document-topic distribution returned by LDA to select the set of documents $D_1, D_2, \ldots D_N$ where $P(T|D)$ is maximized. We call this set the top $N$ documents for $T$.

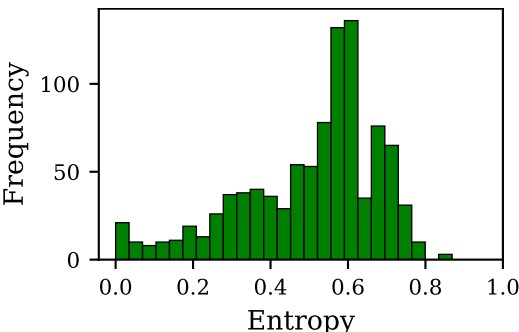
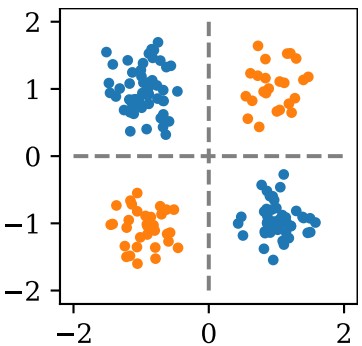

(a) Entropy of visual topics on a suite of image datasets (Appendix A.3. Our topics originate from a diverse set of classes, highlighting the difference between topics and clusters.

(b) Mixture of Related Gaussians. "Words" are generated from orange and blue topics, where opposite quadrants are highly related.

Figure 3: Topics are different than clusters: (a) graphs the class entropy of each visual topic, showing that our topics span multiple classes. (b) exemplifies the relatedness property as a mixture of related Gaussians; there does not exist a clustering that separates the two topics while keeping them intact.

analogous to word clusters in our visual vocabulary. As we well see in the next section, segment clusters are theoretically unable to capture relations of segments.

### 3.2 THE INABILITY OF CLUSTERS TO CAPTURE RELATIONS

In this section, we analyze the difference between clusters and topics from a theoretical point of view. Using the example shown in Figure 3b, we prove that similarity-based clustering techniques cannot capture relatedness.

For example, assume features, or "words", $z_1, z_2, \ldots z_d$ are generated from one of two topics. Each topic is a mixture of two Gaussian distributions. Informally, two features are highly related if they are generated from the same topic. Figure 3b plots a sample of these generated features, where blue points are generated from Topic 1 and orange points are generated from Topic 2. Examples, or "documents", are generated by concatenating a sequence of words, or features from the same topic. Example 1 formalizes this generation procedure as a Mixture of Related Gaussians.

To quantify the relatedness of two features, we use pointwise mutual information (PMI) (Lau et al., 2014; Bianchi et al., 2020). Two words in a topic, or two features generated from the same topic, should have a high PMI. For example, any two blue points in Figure 3b will have a higher PMI than an orange and blue pair. Property 1 formalizes this claim.

**Property 1.** *Two subsets of features $S, S'$ are* related *if their PMI, $\sum_{z_i \in S} \sum_{z_j \in S'} \frac{p(z_i, z_j)}{p(z_i)p(z_j)}$, is high, where $p(z_i, z_j)$ is the probability of observing the feature subsets $z_i$ and $z_j$ within the same example $x$ and $p(z_i)$ is the probability of observing $z_i$ in the entire dataset.*

**Example 1.** *(Mixture of Related Gaussians) Let $x_i \in \mathbb{R}^{2d}, z_{ij} \in \mathbb{R}^2$ be observations from the following generative process:*

- *Let $R_i$ be a Rademacher variable representing the topic, which is drawn from $\{-1, +1\}$ with $\frac{1}{2}$ probability each.*

- *Let $z_{ij} \sim \sum_{S \in \{-1,1\}} p(z|R_i S, S) p(S)$ where $p(z|R_i S, S) = \text{Gaussian}\left([R_i, S], \sigma^2 I\right)$ and $p(S)$ is a Rademacher variable for $j = 1 \ldots d$,*

- *Then, $x_i = (z_{i1}, \ldots, z_{id})$ is the concatenation of all the $z_{ij}$ for $j = 1 \ldots d$.*

The data from Example 1 can be viewed as a mixture of 4 Gaussian distributions with a twist: data from opposite Gaussians are highly correlated. In other words, pairs of features with the same sign are likely to show up together in examples. Conversely, pairs of features with opposite signs are unlikely to appear together in an example. These pairs of opposing quadrants satisfy the *relatedness* property, while adjacent quadrants are not related. From a language perspective, $x_i$ is analogous to a visual document, and $z_{ij}$ is analogous to the $j$th visual word in the $i$th document.

**Theorem 1.** *Let $z_{ij}$ be generated according to Example 1 with $\sigma \leq 0.288$. With probability at least 0.99 there does not exist a clustering of the features $z_{ij}$ that has two clusters containing only related points from each pair of opposing quadrants.*

**Corollary 1.** *A visual topic model with $T = 2$ topics is sufficient to divide the data from Example 1 into two subsets with high relatedness.*

We prove Theorem 1 and Corollary 1 in Appendix A.2. In doing so, we prove that (1) we cannot cluster distinct but related features from opposite quadrants simultaneously for both topics, and (2) a topic model can successfully group all related blue and orange words into two different topics.

We show our visual topics span many classes, demonstrating how they are different from clusters and classes. We theoretically prove that topics discover relationships that clusters cannot represent. Our topics capture themes grounded in relatedness, not similarity, and are thus a complimentary addition to classic unsupervised explanation methods.

## 4 Visual Topic Evaluation

To assess the strength of our visual topics as explanation tools, we evaluate them with well-established metrics in the topic modeling literature (Lau et al., 2014; Abdelrazek et al., 2022):

1. **Human Interpretability**: We conduct a word intrusion test Chang et al. (2009) for our visual topics to evaluate their human-interpretability.
2. **Internal Coherence**: We measure how semantically similar the words within each obtained topic are to maximize topic clarity (Newman et al., 2010).
3. **Relatedness**: We measure how related the words within each topic are. (e.g. "ball" and "jersey" are related, whereas "jersey" and "uniform" are similar.) (Ding et al., 2018)
4. **Diversity**: We measure topic diversity to minimize redundancy (Dieng et al., 2020).

**Experiments.** We apply our visual vocabularies algorithm to a range of image datasets as outlined in Algorithm 1 (see Appendix A.3 for descriptions of included datasets). We then run LDA (Blei et al., 2003) on the resulting documents for each dataset, setting $N_{Topics} = |\text{Num Classes}|/2$. See Appendix A.1 for experimental setup detailing choice of segmentation model $S$, number of clusters $K$, embedding model $F$, and LDA priors.

### 4.1 Human Evaluation

A goal of any good explanation method is human-interpretability (Zhou et al., 2021; Doshi-Velez & Kim, 2017). Linguistic topics have been shown to be understandable to humans Blei et al. (2003), hence their popularity as an explanation tool in NLP. To assess the human-interpretability of our visual topics, we conduct a user study.

A standard way to assess how well linguistic topics match human concepts is through a task known as *word intrusion* (Chang et al., 2009) – users are given a set of words with high probability in Topic A, with the exception of an "intruder", or a word with very low probability in Topic A. Users are then asked to identify this intruder word. If topic A is highly coherent and has some common underlying theme, it should be clear to a user which word is not part of topic A.

**Visual word intrusion.** We design a parallel intrusion task for vision. Given a topic $T$, we first select 4 "exemplar" images $D_1, \ldots D_4$ for that topic, or the images with the largest $P(T|D)$. We highlight all segments $w$ with high membership in that topic, or segments with $P(w|T)$ in the upper quartile. We dim the remaining image so users can focus on the relevant words, while still having the context of the full image to make sense of the unobscured segments. Users are then shown the 4 exemplar images for each topic, along with a randomly chosen intruder image, which is an exemplar document for a different topic. Users are then asked to identify the intruder.

Figure 4 shows an example from this task. We run LDA on the SUN397 dataset using 150 topics, and have a group of 12 volunteer users analyze 25 topics each, with two different users analyzing every subset of 25 topics. Additional information regarding study setup is given in Appendix A.4.

| Image A | Image B | Image C | Image D | Image E |

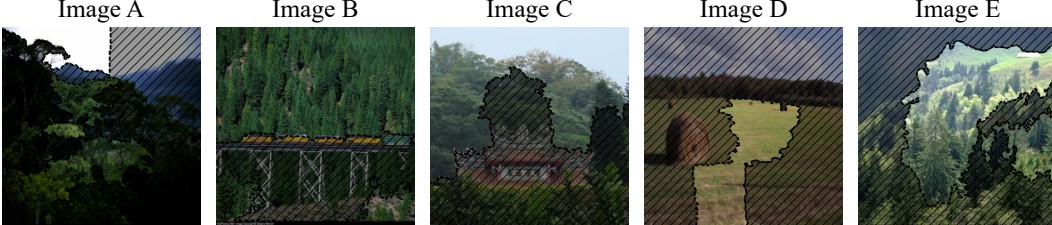

Figure 4: Example of our visual word intrusion task. Users are asked to identify which image they believe is the "imposter", or which image's colored segments belong to a different topic than those of the other four images. In this example, the colored segments in images A, B, C, and E are part of topic that mainly contains clouds, lush forests, and blue skies. Image D, the intruder image, contains segments that belong to a different topic, one that includes grassy fields.

| | **Topic Subset** | **User 1** (% correct) | **User 2** (% correct) | **Agreement** (Cohen's kappa) |
|---|---|---|---|---|
| | 0-24 | 0.88 | 0.96 | 0.891 |
| *Human* | 25-49 | 0.84 | 0.8 | 0.747 |
| *Evaluation* | 50-74 | 0.88 | 0.96 | 0.797 |
| *Results* | 75-99 | 0.92 | 0.96 | 0.896 |
| | 100-124 | 0.72 | 0.76 | 0.689 |
| | 125-150 | 0.92 | 0.92 | 0.947 |
| | **Dataset** | | **Num Topics** | **Overall Performance** |
| *Baseline* | Ours: SUN397 | | 150 | **0.877** |
| *Comparison* | Baseline 1: NYT corpus | | 150 | 0.825 |
| | Baseline 2: Wikipedia | | 150 | 0.840 |

Table 1: Results and baseline comparison for our human evaluation. We show what percent of intruder documents are correctly identified, along with annotator agreement for each pair of users. We compare task performance to two linguistic baselines. Humans find our visual topics to be even more interpretable and coherent than linguistic topics.

We observe exceptionally high performance on this task — on average, users select the correct intruder 87.67% of the time. We also observe high agreement between each pair of users (average Cohen's kappa = 0.828), indicating that our visual topics are human-interpretable. We evaluate against Chang et al. (2009)'s word intrusion task on two text corpora, using an identical topic modeling setup — LDA with 150 topics. Based on user performance, our topics are more coherent to humans than the two linguistic baselines. See Table 1 for full results and baseline comparison.

## 4.2 INTERNAL COHERENCE

Internal coherence, introduced by Newman et al. (2010), measures coherence via pairwise similarity of the top $N$ words $w_1, \ldots w_n$ with highest $P(w|T)$ in each topic. We follow Bianchi et al. (2021) and use centroid similarity to calculate pairwise coherence scores for the top $N$ words in each topic. We define the internal coherence of a topic as follows, where $\hat{v}_i$ corresponds to the centroid of the cluster containing word $i$. Following Bianchi et al. (2021); Grootendorst (2022), we use $N = 10$.

$$IC_{Topic} = \frac{2}{N \times (N-1)} \sum_{i=1}^{N-1} \sum_{j=i+1}^{N} cos(\hat{v}_i, \hat{v}_j) \qquad (3)$$

We compare against two baselines:

- *Text baseline*: we use ViT-GPT2 (Kumar, 2022) to caption each image and run LDA on the generated captions. We use word embeddings (Joulin et al., 2016) for cosine similarity.
- *Shuffled baseline*: we conduct a permutation test Fisher (1936), where we take the top $N$ words from each visual topic and shuffle them across topics. We calculate coherence for the new shuffled topics and report the mean of 100 runs.

| | Aircraft | Birdsnap | Caltech-101 | Caltech-256 | Cars | CIFAR-10 | CIFAR-100 | DTD | Flowers | Food | Pets | SUN397 |
|---|---|---|---|---|---|---|---|---|---|---|---|---|
| *Internal Coherence* | | | | | | | | | | | | |
| **Visual** | **0.86** | **0.91** | **0.84** | **0.85** | **0.79** | **0.95** | **0.99** | **0.74** | **0.87** | **0.93** | **0.85** | **0.87** |
| **Text** | 0.24 | 0.25 | 0.24 | 0.25 | 0.21 | 0.24 | 0.24 | 0.25 | 0.26 | 0.26 | 0.25 | 0.25 |
| **Shuffled** | 0.69 | 0.56 | 0.59 | 0.55 | 0.61 | 0.70 | 0.78 | 0.59 | 0.67 | 0.67 | 0.60 | 0.55 |
| *Topic Relatedness* | | | | | | | | | | | | |
| **Visual** | 0.15 | 0.17 | **0.2** | **0.19** | 0.02 | **0.19** | **0.24** | **0.1** | **0.17** | **0.21** | **0.2** | **0.17** |
| **Text** | **0.22** | **0.19** | 0.1 | 0.08 | **0.14** | 0.12 | 0.09 | 0.08 | 0.06 | 0.1 | 0.05 | 0.1 |
| **Shuffled** | -0.29 | -0.25 | -0.32 | -0.28 | -0.29 | -0.05 | -0.16 | -0.35 | -0.34 | -0.11 | -0.2 | -0.22 |
| *Topic Diversity* | | | | | | | | | | | | |
| **Visual** | **0.76** | **0.83** | **0.88** | **0.89** | **0.77** | **0.97** | **0.87** | **0.99** | **0.92** | **0.89** | **0.97** | **0.89** |
| **Text** | 0.14 | 0.11 | 0.37 | 0.32 | 0.2 | 0.26 | 0.38 | 0.36 | 0.23 | 0.33 | 0.35 | 0.23 |
| **Random** | 0.56 | 0.56 | 0.58 | 0.58 | 0.57 | 0.65 | 0.60 | 0.58 | 0.57 | 0.61 | 0.62 | 0.60 |

Table 2: As is standard in topic modeling literature, we evaluate our visual topics for internal coherence, relatedness, and diversity. We compare against text topics (i.e. LDA on the text captions of each image) and randomly shuffled/sampled topics, where we report the mean of 100 runs.

Table 2 shows the internal coherence of our visual topics and baselines. For all datasets, our visual topics outperform both baselines. These high coherence scores empirically support the findings of our user study — our topics are highly coherent by both human evaluation and automatic metrics.

### 4.3 TOPIC RELATEDNESS

To highlight how visual topics are distinct from segment clusters, we additionally measure relatedness of topics (Ding et al., 2018). The intuition behind this test is that visual words that are not necessarily similar, but co-occur in visual documents (i.e. dogs and grass, boats and lakes, etc.) should still be in the same topic. Relatedness is measured via pointwise mutual information, as discussed in Property 1. We use normalized pointwise mutual information (NPMI) between each pair of words in the top N words. We follow Lau et al. (2014); Grootendorst (2022) and use $N = 10$.

$$R_{Topic} = \frac{2}{N \times (N-1)} \sum_{i=1}^{N-1} \sum_{j=i+1}^{N} \frac{\log \frac{P(w_i, w_j)}{P(w_i)P(w_j)}}{-\log P(w_i, w_j)} \tag{4}$$

We compare against two baselines:

- *Text baseline*: we use ViT-GPT2 (Kumar, 2022) to caption each image and run LDA on the generated captions, using Equation 4 to measure relatedness.
- *Shuffled baseline*: we conduct a permutation test Fisher (1936), where we take the top $N$ words from each visual topic and shuffle them across topics. We calculate the relatedness of these shuffled topics and report the mean of 100 runs.

Table 2 shows the relatedness of our visual topics and baselines. For 9 of the 12 datasets, visual topics are more related than the text topics. For all datasets, our visual topics are more related than the shuffled topics. As discussed in Section 3, the high relatedness of our visual topics indicates we have successfully captured relationships of co-occurring structures in images.

### 4.4 TOPIC DIVERSITY

Dieng et al. (2020) introduce topic diversity as a way to evaluate topic quality. Topic diversity looks at the top $N$ words $w_1, \ldots w_n$ with highest $P(w|T)$ in each topic, and computes the percentage of unique words. A word is considered unique if it is not present in the top $N$ words of any other topics. The intuition behind this metric (ranging from zero to one) is that topic models should

generate diverse topics — a lower diversity score suggests redundant topics, indicating the topics cannot sufficiently disentangle the corpus's themes. A higher score indicates more varied topics. We follow Dieng et al. (2020) and use $N = 25$.

We compare against two baselines:

- *Text baseline*: we use ViT-GPT2 (Kumar, 2022) to caption each image and run LDA on the generated captions. We then calculate diversity.
- *Random baseline*: As shuffling the top $N$ words in each topic will not impact the diversity score, we randomly sample $N$ words from our dataset to create random topics. We calculate diversity of the randomized topics and report the mean of 100 runs.

Table 2 shows the diversity of our visual topics. We outperform both baselines across all datasets. The low diversity of our text baseline indicates that captioning via an image-to-text model loses valuable information needed to create unique and useful topics.

## 5 RELATED WORK

**Dimensionality Reduction.** We first discuss relevant past work in dimensionality reduction methods. Linear dimensionality reduction methods like PCA (Hotelling, 1933) and non-linear dimensionality reduction methods like t-SNE (van der Maaten & Hinton, 2008), MDS (Kruskal, 1964), Isomap (Tenenbaum et al., 2000), UMAP (McInnes et al., 2018), etc. have long been used to find patterns in image data. Cluster memberships of examples can also be used as a form of dimensionality reduction (Jeon, 2001; Park et al., 2003; Berry, 2004). More recent work in this space includes extracting concepts specific to a trained model or class, such as Automatic Concept Extraction (Ghorbani et al., 2019) or Concept Bottlenecks (Koh et al., 2020).

**Unsupervised Interpretability.** Another line of research attempts to explain image datasets via unsupervised methods. Segmentation (Long et al., 2015) divides images into structured sub-parts. Image-to-text models (Vinyals et al., 2015) explain images via captioning, grounding an image in language. Linear probes (Alain & Bengio, 2018; Kornblith et al., 2019) are simple linear classifiers trained on top of existing learned representations and are used to measure the quality of these learned features. Shift explanation techniques, such as those from Kifer et al. (2004); Rabanser et al. (2018); Stein et al. (2023) work towards explaining shifts in data distribution using unsupervised methods. Clustering methods like KMeans (MacQueen, 1967), DBSCAN (Ester et al., 1996), HDBSCAN (Campello et al., 2013), Agglomerative Hierarchical Clustering (Müllner, 2011), Spectral Clustering (Ng et al., 2001), etc. also aim at organizing data using intrinsic structure, which provides unsupervised interpretability.

**Topic Modeling.** Topic modeling is a long-standing field in NLP, and includes fully unsupervised methods such as Latent Dirichlet Allocation (Blei et al., 2003), Latent Semantic Analysis (Deerwester et al., 1990), and Non-Negative Matrix Factorization Lee & Seung (2000). Neural-based topics models, such as BERTopic Grootendorst (2022) and Contexualized Topic Models Bianchi et al. (2020) incorporate pre-trained embeddings in the topic generation process. Topics have been used to compare models (Havaldar et al., 2023) and used for downstream tasks such as summarization, information retrieval, etc. in variety of domains (Jelodar et al., 2019).

## 6 CONCLUSION

In this paper, we proposed visual topic modeling to uncover co-occurrence based relationships in images. We empirically show that topics capture structures that differ from what classic unsupervised learning methods capture for images. We support these results with a theoretical example, which demonstrates that similarity-based clustering is fundamentally unable to model relationships. Our approach generates a visual vocabulary, which can then interface images with topic modeling algorithms from natural language processing.

Topic modeling has long been used to understand text documents in fields such as the social sciences, medicine, and psychology. We hope this paper paves the way for researchers to analogously explain image datasets in these domains.

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

# A  APPENDIX

## A.1  EXPERIMENTAL SETUP

We discuss all details in our visual topic modeling experiments:

**Visual vocabulary construction**   We use SLIC (Achanta et al., 2010) to obtain the segments for each image. We then follow Ghorbani et al. (2019) and crop + resize each segment to get the segment embeddings. We use a vision transformer (Dosovitskiy et al., 2021) to embed these cropped and resized segments. To construct the vocabulary, we use $k$-means clustering and group similar segments together to create visual words. The total number of clusters for each dataset is $25 \times$ Num Classes, following (Salman et al., 2020a).

In our preliminary experiments with SLIC, we experiment with both zeroing (i.e. blacking out) the rest of the image, cropping the segments, and resizing to the max image size. We find that following Ghorbani et al. (2019) generates the best visual topics upon manual inspection, so we follow this procedure for our remaining experiments. We also experiment with the Segment Anything Model (Kirillov et al., 2023), and find that it performs worse than SLIC, upon manual inspection of the resulting topics.

**Topic modeling**   We run LDA implemented using collapsed Gibbs sampling, due to its efficiency and quicker runtime. We set $N_{Topics} = |\text{Num Classes}|/2$ for all datasets. For our LDA prior parameters, we set $\alpha = 0.05$ and $\beta = 0.005$. The standard LDA priors for linguistic topics are $\alpha = 0.1$ and $\beta = 0.01$. Given our visual vocabulary does not follow a Zipfian distribution, like the English vocabulary does, we half the standard priors, as we expect each document to have fewer topics and each topic to have fewer words. For our text baselines, we use the ViT-GPT2 model from HuggingFace to generate the captions for each image. We run LDA for 2,000 iterations to ensure enough time for convergence.

## A.2  PROOFS

**Theorem 1** *Let $z_{ij}$ be generated according to $\sigma \leq 0.288$. With probability at least $0.99$ there does not exist a clustering of the features $z_{ij}$ that has a cluster containing only related points from each pair of opposing quadrants.*

*Proof.* For sake of contradiction, suppose there exists a cluster centroid $\mu_1$ that contains two points $z_1, z_2$ generated from $p(z|1, 1)$ and $p(z| - 1, -1)$, and a second centroid $\mu_2$ that contains two points $z_3, z_4$ generated from $p(z|1, -1)$ and $p(z| - 1, 1)$.

Let $D$ be the Mahalanobis distance with probability $1 - \alpha$ of being within distance $D$ of the mean. Since $\sigma \leq 0.288$, the probability that $z_i$ is at least $D \geq 1$ away from its corresponding centroid is at at most $F\left(\frac{1}{\sigma^2}\right) = F(11.983) = 0.0025 = \alpha$, since $D^2 = \sum_j (z_{ij} - \mu_j)^2/\sigma^2$ follows a $\chi_2^2$ distribution. Then, With probability $p = (1 - \alpha)^4 = 0.99$ (via union bound), all four points $z_i$ are within radius 1 from their respective centroids. This implies that each $z_i$ must lie within one of the four quadrants.

However, note that cluster centroids partition the space into convex partitions. Thus, if $z_1$ and $z_2$ are in the same cluster, then the line connecting them must also be in the same cluster. However, this must also be true of $z_3$ and $z_4$—the line connecting these points must be within the same cluster. Since each of these points exist in opposite quadrants, there must be a point where these two lines intersect that is in the interior of both cluster partitions. However, this is a contradiction, as this point cannot simultaneously be in the interior of two partitions.

$\square$

**Corollary 1** *A visual topic model with $T = 2$ topics is sufficient to divide the data from Example 1 into two subsets with high relatedness.*

*Proof.* Let $v : \mathbb{R}^2 \to \{\pm 1\}^2$ be defined as $v(z_i) = \arg\min_{\mu \in \{\pm 1\}^2} \|z_{ij} - \mu\|_2^2$, which is the hard assignment of each feature subset $z_{ij}$ to the quadrant that $z_{ij}$ resides in. Let $g : \mathbb{R}^{d \times 2} \to \{\pm 1\}^{d \times 2}$

be defined as $g(x_i) = \begin{bmatrix} v(z_{i1}) \\ \vdots \\ v(z_{id}) \end{bmatrix}$ which maps each example $x_i$ to a sequence of words, where each word comes from the vocabulary $\{\pm 1\}^2$. Then, the topic model defined by the topic-term distribution $p(1, 1|T = 1) = p(-1, -1|T = 1) = 0.5$ and $p(-1, 1|T = 2) = p(1, -1|T = 2) = 0.5$ perfectly clusters the examples into two topics with high relatedness. $\square$

## A.3 DATASETS

We follow Salman et al. (2020b) and use the following 12 datasets for our experiments.

| Dataset | Num Images | Num Classes | Content |
| --- | --- | --- | --- |
| Birdsnap Berg et al. (2014) | 40,754 | 500 | North American birds |
| Caltech-101 Fei-Fei et al. (2004) | 8,677 | 101 | Objects |
| Caltech-256 Griffin et al. (2022) | 30,607 | 256 | Objects |
| CIFAR-10 Krizhevsky (2009) | 60,000 | 10 | Objects |
| CIFAR-100 Krizhevsky (2009) | 60,000 | 100 | Objects |
| Describable Textures (DTD) Cimpoi et al. (2014) | 5,640 | 47 | Textured images |
| FGVC Aircraft Maji et al. (2013) | 10,200 | 102 | Aircrafts |
| Food-101 Bossard et al. (2014) | 101,000 | 101 | Food dishes |
| Oxford 102 Flowers Nilsback & Zisserman (2008) | 8,189 | 102 | Flowers |
| Oxford-IIIT Pets Parkhi et al. (2012) | 7,349 | 37 | Cats and dogs |
| SUN397 Xiao et al. (2010) | 39,700 | 397 | Various scenes |
| Stanford Cars Krause et al. (2013) | 16,185 | 397 | Types of cars |

Table 3: Summary of datasets used to generate visual topics.

## A.4 HUMAN EVALUATION

We sourced users to participate in our human evaluation by asking graduate students to volunteer in our study. Our users came from varying academic backgrounds. Of the 12 users, 5 had prior familiarity with topic modeling. As identifying topics requires users to clearly distinguish things like colors and shapes, we additionally required users that users have no visual impairments (e.g. red-green color blindness).

These were the instructions our users received prior to participating in the study:

```
In this study, you will be asked to evaluate the output of a topic
modeling study.  Topic modeling is a technique used to find related
themes in large datasets.  For example, if you were to run a topic
model on Wikipedia articles, an example topic could be words related
to politics, and may contain words like "government", "war", "Iraq",
"Obama", etc.

We have applied a topic modeling algorithm to images, so you will be
asked to evaluate a set of 25 visual topics.  You will see a collection
of 5 images.  4 of these images have segments highlighted that belong
to one visual topic.  1 image has segments highlighted for a different
topic, or is an "intruder" image.  Your task is to look at the set of 5
images and then highlighted segments and identify the intruder.
```

We then showed users an example from the dataset that they were not responsible for annotating, and asked if they understood the task. All users indicated they understood the task after receiving the instructions.

