# OpenReview forum: "Visual Topics via Visual Vocabularies"
_ICLR.cc/2024/Conference — ICLR 2024 Conference Withdrawn Submission_

### Official Review · Reviewer_mBML · 2023-10-17

**Soundness:** 1 poor
**Presentation:** 1 poor
**Contribution:** 1 poor
**Rating:** 1
**Confidence:** 5

**Summary:**

The authors propose a visual topic model that applies topic modeling to images.

**Strengths:**

-

**Weaknesses:**

This paper proposes something as new which exists already. There have been countless works on visual topic modeling, none of them are cited.

The comparison is only with dimensionality reduction methods, not with other topic modeling methods.

Please check for example the following overview article for some example references:
Blei, D., Carin, L., & Dunson, D. (2010). Probabilistic topic models. IEEE signal processing magazine, 27(6), 55-65.

**Questions:**

How is your work different from existing visual topic models?

---

### Official Review · Reviewer_GURR · 2023-11-01

**Soundness:** 2 fair
**Presentation:** 2 fair
**Contribution:** 2 fair
**Rating:** 3
**Confidence:** 4

**Summary:**

This paper proposes visual topic modeling, which is an approach for mapping images into visual words. Thereafter, LDA is used as a topic modeling algorithm to convert the visual words into visual topics. The generated visual topics are evaluated using standard topic modeling metrics and confirm the interpretability of those topics via a human study.

The main intuition behind this paper is to consider images as documents, represented by their visual words. Therefore, a topic modeling algorithm can be used to find relevant topics that are more relevant and interpretable as compared to using clustering algorithms.

The contributions of this paper are as follows:
- a visual topic modeling approach is proposed to convert images into visual words
- visual topics generated by this approach are of a good quality according to topic modeling metrics and highly interpretable via human evaluation

**Strengths:**

- The paper is well written and easy to read
- The proposed approach is described with a lot of details (specifically in the appendix) and is reproducible.

**Weaknesses:**

- The proposed approach for creating a visual vocabulary is very similar to the Bag-of-Visual-Word (BoVW) model, if not the same, and can not be considered a major contribution.

[1] Sivic & A. Zisserman (2003). "Video Google: A Text Retrieval Approach to Object Matching in Videos" (PDF). Proc. of ICCV.
[2] G. Csurka; C. Dance; L.X. Fan; J. Willamowski & C. Bray (2004). "Visual categorization with bags of keypoints". Proc. of ECCV International Workshop on Statistical Learning in Computer Vision.

- There are many previous work using a very similar approach to this paper, i.e. extracting visual words and using topic modeling (LDA) for image annotation [3, 4]. The authors failed to mention these papers in the related work section, and also failed to mention how their approach is different as compared to those.
[3] Topic Modeling of Multimodal Data: An Autoregressive Approach. Yin Zheng, Yu-Jin Zhang, Hugo Larochelle; Proceedings of the IEEE Conference on Computer Vision and Pattern Recognition (CVPR), 2014, pp. 1370-1377
[4] Putthividhy, Duangmanee, Hagai T. Attias, and Srikantan S. Nagarajan. "Topic regression multi-modal latent dirichlet allocation for image annotation." 2010 IEEE Computer Society Conference on Computer Vision and Pattern Recognition. IEEE, 2010.

- A comparison of the proposed visual word to existing visual word models is missing (for instance the Bag-of-Visual-Word model). Also, comparing the different visual topics using LDA on the different visual word extraction models.

**Questions:**

- How the visual words/vocabulary presented in this work is different from prior work?
- Same question for the visual topics?
- What are some applications of the visual topics proposed in this paper?

---

### Official Review · Reviewer_x5q5 · 2023-11-05

**Soundness:** 2 fair
**Presentation:** 2 fair
**Contribution:** 1 poor
**Rating:** 5
**Confidence:** 4

**Summary:**

The idea is to build visual topics for image - in this case  the grouping of similar segments to make visual words - the so called Bag of Word models.

In this way, the paper shares the ambition (and some of the methods, notably LDA) of the recognition literature before neural algorithms became dominant.

But, the athours here are seem also looking a co-occurances (which makes it different from the bulk of those pre-NN algorithms.)

**Strengths:**

I very much like to idea of learning elementary "words" from which "topics" can be built.
And I like the idea of unsupervised learning.

**Weaknesses:**

The topics output are very hard to see (Figure 4).
I cannot see similar segments (visual words) - there is no Figure with, eg a collection of visual words
ie there is no visualisation of the visual vocablaries the authors claim to produce.

I am not sure what I have learned from this paper.
The abstract does not mention co-occurance, but the method (eg Eqn 4) does.

**Questions:**

It is not clear how you learn visual words, or even whether you seek to learn visual objects for noun classes.
This then interefers with reading how you use co-occurance.
And the figures do not help me out at all - they only add to the problems because they are so hard to see.
The conclusion does not match the abstract well.
These factors mean I am unable to fully appreciate your paper or the work behind it.

**Details Of Ethics Concerns:**

none.